# Segregating Suspected Child Maltreatment from Non-Child Maltreatment Injuries: A Population-Based Case-Control Study in Taiwan

**DOI:** 10.3390/ijerph19084591

**Published:** 2022-04-11

**Authors:** Yo-Ting Jin, Chin-Mi Chen, Yao-Ching Huang, Chi-Hsiang Chung, Chien-An Sun, Shi-Hao Huang, Wu-Chien Chien, Gwo-Jang Wu

**Affiliations:** 1Department of Nursing, Fu-Jen Catholic University, New Taipei City 242062, Taiwan; jinyoting@gmail.com (Y.-T.J.); 128135@mail.fju.edu.tw (C.-M.C.); 2School of Nursing, National Taipei University of Nursing & Health Sciences, Taipei 11219, Taiwan; 3Graduate Institute of Medical Sciences, National Defense Medical Center, Taipei 11490, Taiwan; 4Department of Chemical Engineering and Biotechnology, National Taipei University of Technology (Taipei Tech), Taipei 10608, Taiwan; ph870059@gmail.com; 5Department of Medical Research, Tri-Service General Hospital, National Defense Medical Center, Taipei 11490, Taiwan; g694810042@gmil.com (C.-H.C.); hklu2361@gmail.com (S.-H.H.); 6School of Public Health, National Defense Medical Center, Taipei 11490, Taiwan; 7Taiwanese Injury Prevention and Safety Promotion Association, Taipei 11490, Taiwan; 8Big Data Research Center, College of Medicine, Fu-Jen Catholic University, New Taipei City 242062, Taiwan; 040866@mail.fju.edu.tw; 9Department of Public Health, College of Medicine, Fu-Jen Catholic University, New Taipei City 242062, Taiwan; 10Graduate Institute of Life Sciences, National Defense Medical Center, Taipei 11490, Taiwan; 11Obstetrics and Gynecology Department, Tri-Service General Hospital, Taipei 11490, Taiwan

**Keywords:** child maltreatment, child abuse, injuries

## Abstract

Objective: To identify the differential patient characteristics, injury types, and treatment outcomes between hospitalized child abuse and non-child abuse injuries in Taiwan. Methods: Using the data from the National Health Insurance Research Database, we selected a total of 1525 patients under the age of 18 that were diagnosed with child abuse, as well as an additional 6100 patients as a comparison group. Chi-square test, Fisher exact test, and independent samples *t*-test were used to compare the differences between the abused children and the non-abuse-related injured children. The multivariate conditional logistic regression was performed to measure the risk factor of child maltreatment in injured children. Results: Intracranial injury was more frequent in the child abuse group than it was in the non-child abuse group (35.0% vs. 8.2%; *p* < 0.001). Children in the child abuse group tended to stay longer in the hospital and incur higher medical expenses (8.91 days vs. 4.41 days and USD 2564 vs. USD 880, respectively). In multivariate analysis, the Adjusted Odds Ratio (Adjusted OR) of abuse resulting in an injury for children in low-income families is 1.965 times higher than those in non-low-income families (*p* < 0.001). Children living in high urbanization areas had a significantly higher probability of being abused than those living in low urbanization areas (*p* < 0.001). Conclusion: Children under the age of 1 who were hospitalized with severe intracranial injuries are highly at risk for child maltreatment. Moreover, numerous high-risk environmental factors were observed in child abuse cases, including living in urban areas, families with low income, and seasonality, as child maltreatment cases occur more frequently in autumn.

## 1. Introduction

The Centers for Disease Control and Prevention (CDC) defined child maltreatment, also known as child abuse, as any act or series of acts of commission or omission by a parent or other caregiver that results in harm, potential for harm, or threat of harm to a child [1]. Four major categories of maltreatment are commonly recognized: neglect, physical abuse, sexual abuse, and emotional abuse [2]. Child neglect is the most common type of child maltreatment and physical abuse is second to neglect [3,4,5].

Injury is a common physical outcome of child maltreatment. Early detection of subtle injuries in the early stages of child abuse could potentially prevent many fatal or near-fatal abusive events [6,7,8,9,10]. Among non-fatal injuries, bruising injury is the most common injury due to abuse, and one that pediatric providers are often asked to comment on [11]. In comparison, traumatic brain injury is the most common fatal injury in physically abused children, second only to motor vehicle-related injuries in the pediatric age group [12,13].

According to Taiwan’s official data, in 2016 [14] 6031 children were documented as victims of maltreatment, including neglect, physical abuse, sexual abuse, and emotional abuse. Specifically, 37.76% of the victims were physically abused, 30.43% were sexually abused, 19.13% were emotionally abused, and 12.68% were neglected. However, this official statistic contradicts previous studies in that neglect was the most common form of child abuse [3,4]. Taiwan’s fertility rate has dropped sharply, and the population has shown negative growth. However, the number of child abuse cases will not decrease because of fewer children, as it is expected to increase instead of decrease in 2021; it is a national crisis [15].

Although numerous studies have reported on the characteristics of abuse-related injuries, the accuracies of these studies are called into question due to relatively small sample sizes [16].

So far, no studies have compared the characteristics of children admitted to the hospital with abuse-related and non-abuse-related injuries in Chinese societies. Therefore, the aim of this study is to identify the differential patient characteristics, injury types and treatment outcomes between hospitalized child abuse and non-child abuse injuries in Taiwan.

## 2. Materials and Methods

### 2.1. Data Source

The National Health Insurance Research Database (NHIRD) collects nationwide medical data including inpatient, outpatient, and emergency room services, and the law requires that all hospitals and clinics report all medical expenses to the Bureau of National Health Insurance on a monthly basis. Therefore, National Health Insurance (NHI) information can serve as representative empirical data in medical- and health-related research fields [17]. Researchers are required to pass a detailed review by a professional peer review committee before they can use the Taiwan’s NHIRD. The study is a case-control study, used reliable data from the NHIRD, and identified the differential impacts between child abuse and non-child abuse injuries in terms of patient characteristics such as household income, level of care, admission season, geographic region, urbanization level, length of hospital stays, surgical operation, prognosis, and medical costs, as well as the outcomes and injury types in Taiwan.

This study was conducted using the database without patient identifications, thereby conforming to the Declaration of Helsinki. (TSGHIRB number: C202105014).

### 2.2. Subjects

This was a matched case-control study. We selected patients under the age of 18 who were diagnosed with child abuse in accordance with the International Classification of Diseases, Ninth Revision (ICD-9) codes 995.5 Child maltreatment syndrome and E967 Perpetrator of child and adult abuse [18] between January 1997 and December 2013 from the NHIRD as the child abuse group for this study. Category 995.5 includes 995.50 Child abuse, unspecified, 995.51 Child emotional/psychological abuse, 995.52 Child neglect (nutritional), 995.53 Child sexual abuse, 995.54 Child physical abuse, 995.55 Shaken infant syndrome, and 995.59 Other child abuse and neglect; Category E967 Perpetrator of child and adult abuse includes E967.0 By father, stepfather, or boyfriend, E967.1 By other specified person, E967.2 By mother, stepmother, or girlfriend, E967.3 By spouse or partner, E967.4 By child, E967.5 By sibling, E967.6 By grandparent, E967.7 By other relative, E967.8 By non-related caregiver, and E967.9 By unspecified person. The date of first diagnosed child abuse case was treated as the index date. 

The control group was selected from the remaining patients in the NHIRD who were diagnosed with ICD-9-CM codes 800–949 unintentional injury, matched with the cases in terms of gender, age group (<1, 1–2, 3–5, 6–11, and 12–18), and year of the index date. Figure 1 illustrated the criteria of sampling the child abuse group and the control group.

### 2.3. Variable Definitions

The following variables were included in the comparison: household income (USD) (two groups consisting of patients from low-income and non-low-income families), urbanization level of the patient’s residence (in order to correct for the difference in the degree of urbanization between the violent child abuse cases and the areas where the control group lives, the urbanization degree can be defined as the following: Four levels, with 1 being the “most urbanized”, and 4 the “least urbanized”), geographic location (Northern, Central, Eastern, Southern Taiwan, and Outlets islands), admission season, injury types, and outcomes.

Injury types were classified into 21 groups in accordance the ICD-9-CM: 800–804 Fracture of skull, 805–809 Fracture of neck and trunk, 810–819 Fracture of upper limb, 820–829 Fracture of lower limb, 830–839 Dislocation, 840–848 Sprains and strains of joints and adjacent muscles, 850–854 Intracranial injury, 806–869 Internal injury of chest, abdomen, and pelvis, 870–879 Open wound of head, neck, and trunk, 880–887 Open wound of upper limb, 890–897 Open wound of lower limb, 900–904 Injury to blood vessels, 905–909 Poisonings and toxic effects, 910–919 Superficial injury, 920–924 Contusion with intact skin surface, 925–929 Crushing injury, 930–939 Foreign body entering through orifice, 940–949 Burns, 950–957 Injury to nerves and spinal cord, 958–959 Traumatic complications, and 990–999 Others.

Outcomes were measured by level of care (three levels: medical center, regional hospital, and local hospital), catastrophic illness (two groups consisting of patients with/without catastrophic illness, such as cancers, Injury Severity Score ≥ 16, and rare diseases), Charlson Comorbidity Index (CCI), length of hospital stays (days), medical cost (USD), Surgery (with/without), and prognosis (survive/mortality). CCI selects the first five diagnostic codes (ICD-9-CM N-Code), weighs these 5 codes according to 19 disease scoring criteria defined by Charlson, and calculates the total score. The higher the score, the more complications or more severe diagnosis [19].

### 2.4. Statistical Analysis

The SPSS 20.0 statistical software was utilized in this research. Chi-square test, Fisher exact test, and independent samples *t*-test were used to compare the differences between the abused children and the non-abuse-related injured children in demographics, injury type, and outcomes. Furthermore, we calculated the Crude Odds Ratios (crude ORs) and the 95% confidence interval (CI) using the bivariate conditional logistic regression (conditioned on gender, age, and year of the index date) to assess the crude ORs of child abuse in injured children. The multivariate conditional logistic regression was performed to measure the risk factor of child maltreatment in injured children after adjustment for sociodemographic characteristics, outcomes, and injury types. A significance level of α = 0.01 was established to determine the significance of the results.

## 3. Results

Table 1 shows the distribution of demographic characteristics in the child abuse group and the control group. In total 1525 children diagnosed with child abuse were identified and were matched to 6100 controls. Of the total of 7625 children, the mean age was 6.02 years (SD 6.37 years); 60.26% were males. After matching for age group, gender, and year of index date, the two groups were significantly different on household income (USD).

(*p* < 0.001): 5.18% of children in the child abuse group were from low-income families as compared with 3.84% in the control group. More children in the child abuse group lived in northern Taiwan (39.28% vs. 34.29%) and in areas with the highest urbanization level (37.38% vs. 26.07%). Furthermore, children in the child abuse group were hospitalized more frequently in autumn (27.54% vs. 21.39%) and less frequently in winter (23.08% vs. 27.31%) than were their counterparts.

Table 2 shows the distribution of injury types in the child abuse group and the control group. Intracranial injury was more frequent in the child abuse group than it was in the control group (35.02% vs. 8.23%; *p* < 0.001), whereas internal injury of chest, abdomen and pelvis were more frequent in the control group (2.75% vs. 30.26%; *p* < 0.001). Moreover, the rate of the fracture of upper limb for the children in the control group was also significantly higher than that for the children abuse group (21.85% vs. 3.80%; *p* = 0.001).

Table 3 shows the distribution of treatment outcomes in the child abuse group and the control group. Children in the child abuse group were significantly more likely to receive medical attention in hospital centers (47.41% vs. 28.59%), while those in the control group were significantly more likely to receive medical attention in regional hospitals (53.30% vs. 34.16%). In the child abuse group, 3.02% of children had catastrophic illness compared with 1.79% in the control group. In comparing the CCI score, the abused children scored higher than the non-abused children (0.1 vs. 0.02), meaning the number or severity of injury was much higher for the child abuse group. Moreover, children in the child abuse group tended to stay longer in the hospital and incur higher medical expenses (8.91 days vs. 4.41 days and USD 2564 vs. USD 880, respectively). Although children in the child abuse group were less likely to receive surgical operations (23.02%) than those in the control group (42.21%), the mortality rate of children in the child abuse group was significantly higher than those in the control group (5.90% vs. 0.48%).

Table 4 shows the crude and adjusted OR for child abuse among the sampled subjects. The risk of abuse resulting in an injury for children in low-come families is 1.965 times higher than those in non-low-income families (*p* < 0.001). Additionally, a further look at the urbanization level revealed that children living in high urbanization areas had a significantly higher probability of being abused than those living in low urbanization areas (Adjusted OR = 1.560, *p* < 0.001). With respect to CCI score, the higher the score the children had, the higher the probability that they were abused (Adjusted OR = 2.099, *p* < 0.001). The risk of abuse resulting in an injury was significantly higher in autumn than it was in spring (Adjusted OR = 1.545, *p* < 0.001).

Furthermore, we analyzed the adjusted OR of child abuse with respect to injury types. Children with internal injury of chest, abdomen, and pelvis were less likely as a result of abuse (Adjusted OR = 0.121, *p* = 0.001) while children with intracranial injuries were more likely to be abuse victims (Adjusted OR = 1.234, *p* < 0.001) (Table 5).

## 4. Discussion

The results of this study revealed that children with abuse related injuries were more likely to be members of low-income families than those with non-abuse related injuries (5.18% vs. 3.14%). Previous studies have also reported that low-income families may be at greater risk of child maltreatment, which was consistent with our results [20,21,22]. As household income impacts routine medical care and the quality of the caregiving environment, offering adequate economic resources or allowance for parents or caregivers to provide their children with appropriate care may help prevent more children from being abused [21,22].

In terms of urbanization, the percentages of children in the abuse group in the highest urbanized areas and in northern Taiwan were significantly higher than those in the controls group (37.38% vs. 26.07% and 39.28% vs. 34.59%, respectively). This could be because people living in urban areas, especially in developing countries, may experience more stress or depression, and may be associated with higher-risk behaviors (e.g., drug or alcohol abuse), which may increase the risk of child abuse [23,24,25]. Additionally, northern Taiwan is generally more urbanized than any other part of the country. Another observation was that most of the child maltreatment occurred during autumn in Taiwan. This could be linked with seasonal variations in the occurrence of depressive syndromes [26].

Our study has found that intracranial injury was significantly more frequent in the child abuse group than in the control group. Regarding injury types, previous research has indicated that children with abusive head injuries are more likely to die or become more severely incapacitated than children with unintentional head injuries [5]. Thus, it was reasonable to assume that children with abusive head injuries may be more likely to be diagnosed with child abuse. Moreover, we observed that children with internal injury of chest, abdomen, and pelvis were less likely to be diagnosed with child abuse. Although abdominal injury leads to a high risk of in-hospital mortality for children [27], diagnosis of abuse in children with internal abdominal injury was difficult [28]. In addition, the incidence of upper extremity fractures in children in the control group was also significantly higher than that in the child abuse group. Several studies included a total of 154 children who sustained a fracture of the humerus, of whom 30 were classified as abused, 23 had suspected abuse, 100 had fractures resulting from non-abusive injury, and one was involved in a motor vehicle crash, which was similar to our findings [29].

Our study has also found that children in the child abuse group were significantly more likely to receive medical care at a hospital center, while children in the control group were significantly more likely to receive medical care at a regional hospital. Children in the child abuse group had more catastrophic illness than the control group. The number or severity of injuries was much higher in the child abuse group. Children in the child abuse group tended to have longer hospital stays and higher medical costs. Although children in the child abuse group were less likely to undergo surgery than the control group, the child abuse group had a significantly higher mortality rate than the control group.

Children from low-income households are at greater risk of injury from abuse than children from non-low-income households. Children living in highly urbanized areas are significantly more likely to be abused than children living in less urbanized areas. In autumn, children at the risk of injury from abuse is significantly higher than in spring, children with internal chest, abdominal, and pelvic injuries were less likely to be abused, while children with intracranial injuries were more likely to be victims of abuse. Regarding treatment outcomes, children in the abuse group required significantly longer hospital stays and suffered greater risk of fatality compared with those in the control group. In addition, abused children scored significantly higher on injury severity and experienced more catastrophic illness. Generally speaking, serious injuries require more comprehensive care at an advanced medical facility. Hence, nearly half of the abused children (47.41%) were treated at medical centers as opposed to regional and local hospitals in Taiwan. Therefore, the extended length of hospital stays, and the level of care were also associated with the severity of injury and medical costs. Our findings indicated that the average medical costs in the child abuse group were 2.91 times higher than those in the control group. Furthermore, while it appears that the more severe the injury, the higher the likelihood it would require surgical interventions, previous research has reported only one in five unconscious children suffering from head injury requires surgical operations [30]. Our study is consistent with this finding such that children in the child abuse group (23.02%) required fewer surgical interventions than those in the control group (42.21%). This is because while the rate of intracranial injury in the child abuse group was significantly higher than that of the control group (35.02% vs. 8.23%), the severity of the majority of these injuries did not warrant surgeries.

Finally, there were several limitations to this study. First, the study was limited to the data available in the health insurance database. We were unable to consider certain other factors that could also influence child maltreatment risk such as parent–child relationship, marital status, educational level, and religious beliefs. Second, the National Health Insurance database did not provide clinical biochemistry data, the Glasgow Coma Scale [31], or abbreviated injury severity scores [32]. Therefore, we used medical-related factors (e.g., length of hospital stays and medical costs) as indicators of the severity of injury. Third, we were unable to identify and interact with the patients directly to obtain additional information such as mental status due to privacy concerns and protocols. Finally, this study used inpatient data exclusively and we could not obtain sufficient information on cases where the injuries were minor and did not require care, or the patient received only outpatient/emergency care. Therefore, the results of this study were biased toward more severe cases of injury.

## 5. Conclusions

Our study found that children from low-income households, living in highly urbanized areas, and in autumn, are at greater risk of injury from abuse than children from non-low-income households. Children with internal chest, abdominal, and pelvic injuries are less likely to be abused, while children with intracranial injuries are more likely to be victims of abuse. Moreover, our study identified and discussed numerous high-risk environmental factors observed in child abuse cases, including living in urban areas, families with low-come, and seasonality as child maltreatment cases occur more frequently in autumn. These findings will help develop new protocols and diagnostic criteria so that physicians are able to identify and report child abuse in a timelier manner.

This issue is of great significance and it is important to have the latest and most accurate information. Future studies should investigate if anything has changed during the observational period (e.g., comparing every ten years) from 2013 to 2022, including changes in awareness, relevant organizations, child support, etc.

## Figures and Tables

**Figure 1 ijerph-19-04591-f001:**
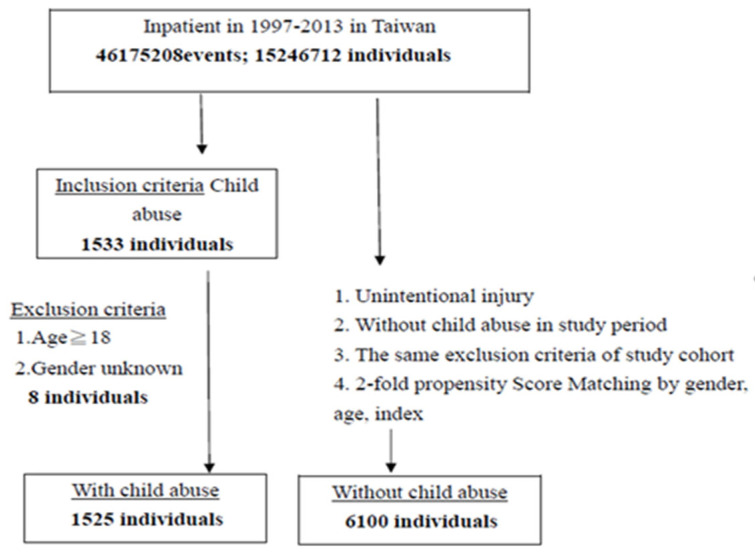
The flowchart of study sample selection from National Health Insurance Research Database in Taiwan.

**Table 1 ijerph-19-04591-t001:** Demographic characteristics of children with child abuse and controls (*n* = 7625).

Variables	Children with Child Abuse		Controls		*p* Value
	*n*	%	*n*	%	
Total	1525	20.00	6100	80.00	
Gender					0.999
Male	919	60.26	3676	60.26	
Female	606	39.74	2424	39.74	
Age (years)					0.999
<1	562	36.85	2248	36.85	
1–2	207	13.57	828	13.57	
3–5	186	12.20	744	12.20	
6–11	166	10.89	664	10.89	
12–18	404	26.49	1616	26.49	
Household income (USD)					<0.001
Without low-income	1446	94.82	5866	96.16	
With low-income	79	5.18	234	3.84	
Geographic region					<0.001
Northern	599	39.28	2110	34.59	
Middle	476	31.21	1974	32.36	
Southern	392	25.70	1535	25.16	
Eastern	56	3.67	448	7.34	
Outlets islands	2	0.13	33	0.54	
Urbanization level					<0.001
1 (most urbanized)	570	37.38	1590	26.07	
2	688	45.11	2751	45.10	
3	79	5.18	603	9.89	
4 (least urbanized)	188	12.33	1156	18.95	
Admission season					<0.001
Spring (March–May)	366	24.00	1685	27.62	
Summer (June–August)	387	25.38	1444	23.67	
Autumn (September–November)	420	27.54	1305	21.39	
Winter (December–February)	352	23.08	1666	27.31	

*p*: Chi-square/Fisher exact test on category variables and *t*-test on continuous variables.

**Table 2 ijerph-19-04591-t002:** Comparison of injury types between children with child abuse and controls.

Injury Types	Children with Child Abuse		Controls		*p* Value
	*n*	%	*n*	%	
Fracture of skull	62	4.07	252	4.13	0.983
Fracture of neck and trunk	5	0.33	49	0.80	0.908
Fracture of upper limb	58	3.80	1333	21.85	0.001
Fracture of lower limb	41	2.69	506	8.30	0.199
Dislocation	1	0.07	77	1.26	0.971
Sprains and strains of joints and adjacent muscles	2	0.13	153	2.51	0.830
Intracranial injury	534	35.02	502	8.23	<0.001
Internal injury of chest, abdomen and pelvis	42	2.75	1846	30.26	<0.001
Open wound of head, neck and trunk	31	2.03	216	3.54	0.622
Open wound of upper limb	9	0.59	128	2.10	0.755
Open wound of lower limb	9	0.59	139	2.28	0.736
Injury to blood vessels	0	0.00	8	0.13	0.594
Poisonings and toxic effects	4	0.26	168	2.75	0.761
Superficial injury	11	0.72	102	1.67	0.810
Contusion with intact skin surface	79	5.18	114	1.87	0.201
Crushing injury	4	0.26	44	0.72	0.915
Foreign body entering through orifice	4	0.26	86	1.41	0.846
Burns	19	1.25	237	3.89	0.557
Injury to nerves and spinal cord	5	0.33	32	0.52	0.955
Traumatic complications	36	2.36	55	0.90	0.573
Others	569	37.31	53	0.87	<0.001

*p*: Chi-square/Fisher exact test on category variables and *t*-test on continuous variables.

**Table 3 ijerph-19-04591-t003:** Outcomes of children with child abuse and controls.

Variables	Children with Child Abuse		Controls		*p* Value
	*n*	%	*n*	%	
Total	1525	20.00	6100	80.00	
Level of care					<0.001
Hospital center	723	47.41	1744	28.59	
Regional hospital	521	34.16	3251	53.30	
Local hospital	281	18.43	1105	18.11	
Catastrophic illness					<0.001
Without	1479	96.98	5991	98.21	
With	46	3.02	109	1.79	
Charlson comorbidity index (CCI)	0.10 ± 0.40		0.02 ± 0.17		<0.001
length of hospital stays	8.91 ± 10.96		4.41 ± 4.41		<0.001
Medical cost (USD)	2563.68 ± 4658.38		880.27 ± 1747.04		<0.001
Surgical operation					<0.001
Without	1174	76.98	3525	57.79	
With	351	23.02	2575	42.21	
Prognosis					<0.001
Survive	1435	94.10	6071	99.52	
Mortality	90	5.90	29	0.48	

*p*: Chi-square/Fisher exact test on category variables and *t*-test on continuous variables.

**Table 4 ijerph-19-04591-t004:** Risk factors associated with child abuse based on conditional logistic regression.

Variables	Bivariate Model			Multivariate Model		
	Crude OR	95% CI	*p* Value	Adjusted OR	95% CI	*p* Value
Low-income	1.370	[1.054–1.779]	0.018	1.965	[1.483–2.604]	<0.001
Urbanization level						
1 (most urbanized)	2.204	[1.839–2.643]	<0.001	1.560	[1.242–1.96]	<0.001
2	1.538	[1.201–1.832]	<0.001	1.285	[0.956–1.562]	0.072
3	0.806	[0.608–1.067]	0.131	0.767	[0.57–1.032]	0.080
4 (least urbanized)	Reference			Reference		
Catastrophic illness						
Without	Reference			Reference		
With	1.708	[1.206–2.472]	0.003	1.385	[0.922–2.079]	0.117
CCI	3.002	[2.385–3.780]	<0.001	2.099	[1.65–2.672]	<0.001
Level of care						
Hospital center	1.630	[1.393–1.904]	<0.001	1.025	[0.653–1.872]	0.351
Regional hospital	0.630	[0.537–1.047]	0.074	0.701	[0.378–1.450]	0.154
Local hospital	Reference			Reference		
Surgical operation						
Without	Reference			Reference		
With	0.680	[0.364–0.912]	0.005	0.761	[0.196–1.356]	0.297
Admission Season						
Spring	Reference			Reference		
Summer	1.234	[1.052–1.447]	0.010	1.279	[0.972–1.524]	0.067
Autumn	1.482	[1.266–1.725]	<0.001	1.545	[1.296–1.84.]	<0.001
Winter	0.974	[0.828–1.143]	0.737	0.928	[0.777–1.109]	0.413

Adjusted OR = Adjusted odds ratio: Adjusted variables listed in the table, CI = confidence interval.

**Table 5 ijerph-19-04591-t005:** Crude and adjusted odd ratios of child abuse for injury types based conditional logistic regression.

Injury Types	Bivariate Model			Multivariate Model		
	Crude OR	95%CI	*p* Value	Adjusted OR	95%CI	*p* Value
Fracture of skull	0.256	[0.121–1.597]	0.982	0.298	[0.121–1.597]	0.982
Fracture of neck and trunk	0.125	[0.059–1.265]	0.886	0.198	[0.035–1.988]	0.886
Fracture of upper limb	0.449	[0.121–0.795]	<0.001	0.896	[0.772–0.995]	0.025
Fracture of lower limb	0.812	[0.254–1.950]	0.195	1.454	[0.898–2.982]	0.232
Dislocation	0.013	[0.002–5.454]	0.965	0.124	[0.012–4.955]	0.772
Sprains and strains of joints and adjacent muscles	0.019	[0.003–3.454]	0.565	0.059	[0.026–3.495]	0.542
Intracranial injury	1.067	[1.011–1.564]	<0.001	1.234	[1.025–1.670]	<0.001
Internal injury of chest, abdomen and pelvis	0.023	[0.002–0.124]	<0.001	0.121	[0.029–0.568]	0.001
Open wound of head, neck and trunk	0.111	[0.029–1.590]	0.645	0.098	[0.033–1.565]	0.644
Open wound of upper limb	0.076	[0.003–2.950]	0.765	0.121	[0.034–1.998]	0.795
Open wound of lower limb	0.065	[0.012–1.986]	0.798	0.055	[0.011–1.749]	0.897
Poisonings and toxic effects	0.024	[0.002–1.465]	0.464	0.101	[0.044–1.989]	0.511
Superficial injury	0.178	[0.022–1.596]	0.765	0.199	[0.102–2.995]	0.564
Contusion with intact skin surface	0.693	[0.465–3.454]	0.295	0.795	[0.345–3.495]	0.265
Crushing injury	0.091	[0.002–2.986]	0.911	0.895	[0.154–2.412]	0.881
Foreign body entering through orifice	0.047	[0.014–4.562]	0.842	0.124	[0.022–4.431]	0.701
Burns	0.081	[0.025–1.113]	0.541	0.894	[0.131–1.234]	0.454
Injury to nerves and spinal cord	0.156	[0.104–4.560]	0.951	0.442	[0.265–4.65]	0.851
Traumatic complications	0.656	[0.557–2.982]	0.575	1.021	[0.598–2.988]	0.568

Adjusted OR = Adjusted odds ratio: Adjusted variables listed in the table, CI = confidence interval.

## Data Availability

Data are available from the NHIRD published by the Taiwan NHI administration. Because of legal restrictions imposed by the government of Taiwan concerning the “Personal Information Protection Act”, data cannot be made publicly available. Requests for data can be sent as a formal proposal to the NHIRD (http://www.mohw.gov.tw/cht/DOS/DM1.aspx?f_list_no=8120 (accessed on 13 October 2021)).

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
