# Peer review of "Segregating Suspected Child Maltreatment from Non-Child Maltreatment Injuries: A Population-Based Case-Control Study in Taiwan"

_ijerph, 2022, doi:10.3390/ijerph19084591_

Round 1
Reviewer 1 Report
The article addresses an important issue, and can add information helpful for clinicians and policy makers.
I have some suggestions regarding
- Introduction: make it more concise, focusing just on the relevant aspects to introduce your research (few words for the background, previous research in the area, and what is missing, what you did to fix that). All these elements are present in your manuscript, just try to focus on them, avoiding too many details.
- Aim of the study: The aim of the study is to identify the differential patient characteristics, injury types and treatment outcomes between child abuse and non-child abuse injuries in Taiwan. I would avoid the referral to the use of the national database describing your aim (more pertinent to Methods)
-Methods: reformulate the last sentence of the Introduction, saying here you used the NHIRD. Clarify again here that your study is a case-control study. Create a specific paragraph for Ethical approval, making it explicit (professional peer review?).
-Discussion: Introduce also the part about patient characteristics as you did for the other ones (Regarding patient characteristics, etc.) in the way it is clear for readers. Use always hypothetical structure while discussing your hypothesis (e.g., This is because people 246 living in urban areas, especially in developing countries, may experience more stress or 247 depression, and are associated with higher risky behaviorschange it to This could be because); this will be safer and more correct, especially speaking of such a delicate issue.
- Spelling and grammar: minor typos regarding punctuation.
Reviewer 2 Report
Major issues:
#1. About income, was this Taiwan dollars? US dollars?
#2. The word “urbanization” in unclear. Please add more explanation.
#3. At the beginning of Discussion, please state the most important findings. The first or other is not a big issue here.
#4. Male >female was not significant in this study. You cannot state the gender difference in Discussion.
#5. The authors need to majorly revise Discussion. This part is scattered and not related the results.
The authors are allowed to state the statistically significant factors and then please discuss why.
#6. Again, the authors suddenly state irrelevant factors in Conclusion. Was “Remaining hidden” proved in this study?
#7. What was the top priority in this study? Firstly, you prioritize the factors that were significant and then discuss them. All factors that were not significant in the study were occasionally seen and discussed. This confuses readers and does not sounds scientific.
Minor issues:
#1. At the end of Introduction, “2” should be erased.
Reviewer 3 Report
no abbreviations in the abstract (OR)
Line 103: 2.?
Use the term patients instead of subjects in the entire manuscript.
Delete comma in numbers in the result section.
Who was the most often identified aggressor according to the age groups?
What were the “suspicious location of the injuries and cause of the injuries” in this study helping to identify maltreated children?
This issue is of great importance nevertheless it is simultaneously important to be at the latest and most accurate information. The results end in with the year 2013 almost 10 years ago. A further work up until 2020 is essential to be accurate and really identify the actual problems even if they might not have changed it needs to be confirmed. I really hope this problem can be solved because this work is of such a great importance but from a scientific view this needs to be addressed.
Further it would be interesting if anything has changed during the observational period (example comparing every ten years). Awareness, organisations, child support in any way etc.
Round 2
Reviewer 2 Report
The authors sincerely revised the manuscript. This paper would be meaningful for the individuals with abuse problems.
Reviewer 3 Report
some changes were made. the substantial ones from my point of view were not sufficiently integrated into the manuscript therefore there is not much to revise. The decision is up to the editor.